# Role of ruscogenin extracted from *Radix Ophiopogon Japonicus* in antagonizing 5-hydroxytryptamine and dopamine receptors through computational screening

**Suya Ma**, **Yongmei Liu** *

Guang'anmen Hospital, China Academy of Chinese Medicine Sciences, Beijing, China

* lymsd@163.com

**Data Availability Statement:** The data is available at the following DOI link: https://doi.org/10.7910/DVN/SQYQSY.

## Abstract

The 5-hydroxytryptamine (5-HT) and dopamine (DA) receptors have emerged as significant targets for therapeutic intervention in psychiatric disorders. Currently, the efficacy of psychiatric drugs is limited by challenges in achieving desired outcomes, the occurrence of adverse effects, dependence, and withdrawal reactions. Consequently, there is a pressing need for the development of safe and effective therapeutic agents for psychiatric disorders. To explore the potential effects of natural product extracts as therapeutic agents for psychiatric disorders, 73 active ingredients from natural medicine extracts were screened to identify potential inhibitors of the serotonin 2A receptor (5-HT2AR) and dopamine D2 receptor (DRD2) using computerized virtual molecular docking. The most effective inhibitor of 5-HT2AR and DRD2 among these natural extracts was then evaluated for its drug-like properties using ADMET analysis, and its mechanisms of antagonism on DRD2 and 5-HT2AR were studied through molecular dynamics simulation. Risperidone was used as a positive control drug. The results showed that ruscogenin (RUS) was the most effective inhibitor of 5-HT2AR and DRD2, possessing favorable drug-like properties (most values of ADMET analysis were within the optimal range). When compared to risperidone, RUS exhibited more stable root mean square deviation (RMSD) plots, lower root mean square fluctuation (RMSF) values from residues 50 to 260, stronger hydrogen bonding interactions, higher compactness, a smaller solvent-accessible surface area (SASA) value, and lower binding free energy (-43.81 kcal/mol vs. -35.68 kcal/mol). RUS also demonstrated inhibitory effects on DRD2, as indicated by stable RMSD plots, low RMSF values from residues 50 to 250, strong hydrogen bonding interactions, high compactness, a small SASA value, and low binding free energy (-35.00 kcal/mol). Consequently, this research suggests that RUS, a natural pharmaceutical extract, is a promising candidate for further validation through clinical studies, representing a potential development of a therapeutic agent targeting psychiatric disorders.

**Funding:** This research was funded by National Natural Science Foundation of China (Key Program) (NO. 82230124).

**Competing interests:** The authors declare no conflicts of interest.

## Introduction

Natural products offer a promising avenue for identifying scaffolds with a wide range of structural diversity and biological activities, which can be utilized directly or serve as a foundation for optimizing novel pharmaceuticals [1, 2]. The bioactive compounds derived from natural products offer additional advantages, including nutritional enrichment, widespread availability, cost-effectiveness, therapeutic potency, and the capacity for multi-target action through interconnected causal pathways, all while minimizing adverse effects [3]. Recently, extensive research has been conducted on the potential use of antipsychotic natural products. According to the available data, natural products used in traditional medicine can be significantly effective in reducing depression, alleviating depressive symptoms, and improving patients' performance [4]. It is evident that the extraction of bioactive compounds from natural products holds significant medicinal value in diseases caused by neurotransmitter abnormalities [5–7].

Dopamine (DA), a monoamine neurotransmitter and hormone, functions in many aspects of human and animal behavior, such as aggression, hallucinations, delusions, depression, emotion, and motor control [8]. Abnormalities in the functioning of the DA system can lead to different diseases, such as schizophrenia, Parkinson's disease, and attention deficit hyperactivity disorder [9]. DA receptors, which have five subtypes, belong to the G protein-coupled receptor family. Dopamine D2 receptor (DRD2), the second most abundant dopaminergic receptor, is mainly expressed in the olfactory bulb, nucleus accumbens, substantia nigra, and ventral tegmental area [9]. DRD2 is a drug target used in the treatment of numerous pathologies, including hypertension, Parkinson's disease, schizophrenia, and other neuropsychiatric disorders [10]. Another important neuroregulator, 5-hydroxytryptamine (5-HT), also operates in numerous aspects of human and animal behavior, including aggression, hallucinations, delusions, depression, anxiety, and appetite. The dysfunction of 5-HT system is closely associated with various pathophysiological conditions of the central nervous system, such as schizophrenia, depression, and anxiety, etc. [11]. Selective serotonin 2A receptor (5-HT2AR) antagonists are among a large number of drugs being studied for psychiatric disorders, and continue to demonstrate promise [12]. Therefore, DRD2 and 5-HT2AR are important pharmaceutical targets involved in signaling pathways underlying various neurological and psychiatric functions and dysfunctions.

To conduct preclinical studies targeting DRD2 and 5-HT2AR with natural product extracts, molecular dynamics (MD) simulation, a subfield of computational biology, is a rapid, efficient, and accurate technique. MD simulation can study the structure of drug target proteins and various other properties from a more microscopic perspective (molecular level) to achieve efficient and accurate virtual screening. Martin Karplus, Michael Levitt, and Arieh Warshel won the Nobel Prize in Chemistry in 2013 for their contributions to computer models of complex chemical systems [13]. MD simulation can make up for the shortcomings of experimental methods, overcoming the limitations of blind, time-consuming, and laborious experiments. It can explain experimental phenomena and provide predictions and guidance for future experiments. Additionally, it allows for real-time observation of the conformational binding process and the acquisition of information regarding conformational changes. Furthermore, it describes and explains the properties of biomacromolecules at the microscopic and atomic levels. MD simulation plays an increasingly important role in the research of biology, pharmacy, chemistry, materials science, and more.

For antipsychotic drugs, although effective in managing the symptoms of psychiatric disorders to a certain extent, they are not without a repertoire of serious side effects [14]. There is a need for better therapies to treat psychiatric disorders, particularly in identifying and targeting DA or 5-HT receptors [14, 15]. In this study, in order to discover effective, safe natural product

extracts for the treatment of psychobehavioral symptoms and other disorders associated with abnormal DA and 5-HT, the strong inhibitory effects of ruscogenin (RUS) on the 5-HT2AR and DRD2 were screened out through virtual screening. What's more, RUS has been shown to possess the ability to exert anti-anxiety effects [16]. Then MD simulation was used to explore the pharmacology mechanism of RUS in inhibiting 5-HT2AR and DRD2, which may have potential clinical uses in diseases associated with abnormal DA and 5-HT.

## Materials and methods

### Virtual screening of active ingredients

The AutoDock Vina 1.2.0 [17] was used to conduct virtual screening of natural medicine extracts for 5-HT2A and DA2 receptor inhibitors. The crystal structures of DRD2 and 5HT2AR were downloaded from the PDB database (https://www.rcsb.org/). The ZINC database (https://zinc.docking.org/) and the TCMSP database (https://old.tcmsp-e.com/tcmsp. php) were utilized to obtain the molecular structures of the natural medicine extracts. PyMol [18] was used to remove the ligands and water molecules from the crystal structures of DRD2 and 5HT2AR, and then the crystal structures were saved as .pdb files by PyMol. AutoDock Vina 1.2.0 [17] was used to add hydrogen atoms and calculate the charges of the DRD2 and 5HT2AR receptors and their respective ligands, and then they were saved in .pdbqt format. Prior to selecting of DRD2 and 5HT2AR receptors and their respective ligands, hydrogen atoms were fully added. The method employed for this process utilized the 'withBondOrder' option. Charges were calculated using the empirical Gasteiger Charges [19]. The original crystal structure of DRD2 and 5HT2AR were referred to in order to set the binding pocket. For DRD2, the binding pocket was defined by the side chains of helices III, V and VI (Fig 2 a1, b1) [20]. For 5HT2AR, the bottom hydrophobic cleft was selected as its bonding pocket (Fig 3 c1, d1) [21]. Given that flexible docking has been observed to be less accurate and more computationally intensive compared to semi-flexible docking [22], we opted for the semi-flexible docking approach, where the receptor is considered to have a rigid structure and the ligand is treated as a flexible structure. A torsion tree was utilized to represent both the fixed and variable segments within the ligand.

### ADMET analysis of RUS

The ZINC database (https://zinc.docking.org/) was utilized to acquire the molecule SMILES of RUS. Additionally, ASMETlab 3.0 [23] (https://admetlab3.scbdd.com) was employed to predict the ADMET (absorption, distribution, metabolism, excretion and toxicity) predicties of RUS.

### Determine the molecular force field and other parameters

Select the structure with the lowest binding energy of AutoDock Vina as the initial structure of the protein-ligand complex (including 5-HT2AR-RUS, 5-HT2AR-risperidone, DRD2-RUS, and DRD2-risperidone). Amber 22.0 [24] was used to perform molecular simulations. Open Babel was employed to hydrogenate the ligands [25]. The antechamber and parmchk2 were used to add the Bond-Charge Correction (bcc) charges and parameters to the ligands [26] and save them as .frcmod parameter files. The tleap program was used to generate .lib files for the ligands [27]. The pdb4amber program was utilized to remove the non-protein residues and water molecules from the protein structure, add the missing heavy atoms for the standard amino acid residues, remove the hydrogen atoms of the amino acid residues, and format it into a .pdb structural file [27]. The tleap module was employed to construct the initial system; the leaprc.protein.ff99SB force field was selected for proteins, and the leaprc.gaff force field

was selected for the ligands. We utilized leaprc.protein.ff99SB to load all the libraries containing the parameters of the AMBER force field ff99SB. This ff99SB parameterization offers a well-calibrated set for simulating proteins, effectively capturing the equilibrium of secondary structural elements [28]. The leaprc.gaff was used to load all the libraries containing the parameters of the Generalized Amber Force Field (GAFF) force field. The GAFF is crafted for seamless integration with established Amber force fields for biomolecules such as proteins and nucleic acids. It encompasses a broad parameter set accommodating a wide array of organic and pharmaceutical compounds consisting of elements like hydrogen, carbon, nitrogen, oxygen, sulfur, phosphorus, and the halogen group [29]. Considering the compatibility of the TIP3P water model with the GAFF force field, the TIP3P water box was selected as the solvent environment for the simulation box [30]. During the simulation, periodic boundary conditions were used to prevent edge effects. The gap between the solute surface and the box is fixed at 12 Å [31]. Then, a suitable amount of $Na^+$ or $Cl^-$ was added to the system for neutralization. For the 5-HT2AR, 1 $Cl^-$ ion was added to neutralize the system for both RUS and risperidone. For the DRD2, 17 $Cl^-$ ions were added to neutralize the system for both RUS and risperidone. The SHAKE algorithm is used in MD simulations to handle the internal constraints of molecules, especially suitable for dealing with hydrogen atoms. Due to their smaller mass, hydrogen atoms vibrate at a faster rate than other atoms in simulations, which may lead to numerical instability. The SHAKE algorithm reduces the computational load and improves the stability and accuracy of the simulation by constraining the distances between hydrogen atoms and other atoms to remain at the predefined equilibrium distances [32, 33]. In this study, all bonds related to hydrogen atoms were restricted by the SHAKE algorithm. The Particle Mesh Ewald algorithm was applied to deal with the non-bonded electrostatic interactions.

## Optimize the initial structure and conduct equilibrium simulation

After the system was initially constructed, a systematic computational protocol was employed. Mobile ligand molecules underwent energy minimization using the gradient steepest descent algorithm, consisting of 1,000 steps segmented into 500 steepest descent and 500 conjugate gradient method steps. This was followed by a 15 ps MD simulation under NVT conditions at 300 K for initial relaxation. For the 5HT2AR and DRD2 receptor molecules, a more extensive 3,000-step energy minimization was conducted, utilizing the steepest descent method for the first 1,000 steps and the conjugate gradient method for the remaining 2,000 steps, followed by a 10 ps MD simulation under NPT conditions at 300 K for initial relaxation. Subsequently, a 10 ps MD simulation focused on the relaxation of non-backbone atoms within the NPT ensemble, complemented by an additional 10 ps simulation for unrestrained relaxation. Post-equilibration, a 40 ns MD simulation sampling was performed under NPT conditions at 300 K and 1 bar pressure [34]. The simulation parameters included a sampling frequency of 1,000 (nscm = 1,000), a trajectory output interval of 1,000 (ntwx = 1,000), a print frequency of 500 (ntpr = 500), a restart frequency of 1,000 (ntwr = 1,000), and a cutoff distance of 9.0 Å (cut = 9.0). Root mean square deviation (RMSD, for structural stability), root mean square fluctuation (RMSF), gyration radius (Rg), solvent-accessible surface area (SASA), and hydrogen bonding (for structural compactness) were calculated using the CPPTRAJ module and presented using the xmgrace program. Trajectories were visualized using VMD [35].

## Calculating binding free energy

GROMACS 2020.1 [36] and gmx_MMPBSA [37] were applied to estimate the binding free energy of protein-ligand interactions via the method of Molecular Mechanics Generalized Born Surface Area (MM-GBSA). Precisely identifying the residues that significantly influence

the thermodynamic stability of molecular interactions is essential for the analysis of binding energy. The alanine scanning mutagenesis technique systematically incorporates alanine into each residue of a protein under investigation, enabling the assessment of the energetic contribution of each side chain to the formation of interactions. This approach provides profound insights into understanding the function and structure of proteins [38]. In this study, the gmx_MMPBSA_ana program [37] was employed to conduct alanine scanning and residue decomposition, which could analyze the contributions of diverse residues and regions to the total binding free energy. The defining secondary structure of proteins (DSSP) was calculated by the CPPTRAJ modules after the analysis of residue decomposition.

## Results

### Virtual screening of active ingredients

The AutoDock Vina 1.2.0 [17] was used to conduct virtual screening of active ingredients in natural medicines. The results of virtual screening of the active ingredients with 5-HT2AR (PDB ID: 6A93) and DRD2 (PDB ID: 6CM4) showed that RUS ($C_{27}H_{42}O_4$) was the best inhibitor, demonstrating the highest binding affinity with 5HT2AR (-10.5 kcal/mol) and (DRD2: -8.85 kcal/mol) among other natural extracts (Table 1). The binding affinities of risperidone with 5-HT2AR and DRD2 using AutoDock Vina were -10.8 kcal/mol and -10.32 kcal/mol respectively. The screening results are shown in Table 1, and the docking structures are presented in supporting information S1 Fig and S2–29 Figs in S1 File.

### ADMET analysis of RUS

ADMET analysis typically occurs post-target potency confirmation for drug candidates. With ADMET-related issues accounting for an estimated 50% of drug development failures in the 1990s, this evaluation is crucial for translating chemicals into viable drugs [39]. RUS, a candidate inhibitor of 5HT2AR and DRD2, had its ADMET profiling evaluated by ASMETlab 3.0 [23]. The SMILES of RUS (ID: ZINC8234225) is C[C@H]1[C@H]2[C@H](C[C@H]3[C@@H]4CC = C5C[C@@H](O)C[C@@H](O)[C@]5(C)[C@H]4CC[C@@]32C)O[C@]12CC[C@@H](C)CO2. The molecular structure of RUS is shown in Fig 1A. Liposolubility is an important parameter of small molecules in medicinal chemistry, typically represented by log P and log D7.4. Log P refers to the logarithm of the n-octanol/water distribution coefficient. High log P compounds have poor water solubility and strong liposolubility. The log P value of RUS exceeded the upper limit (log P = 5.044, Table 2 and Fig 1B), indicating that it has good liposolubility and can easily penetrate the cellular membrane structure. Log D7.4 refers to the logarithm of the n-octanol/water distribution coefficient at pH = 7.4. High log D7.4 compounds represent strong liposolubility. The log D7.4 value of RUS also exceeded the upper limit (log D7.4 = 4.272, Table 2 and Fig 1), indicating that it has good liposolubility and can easily penetrate the cellular membrane structure. Solubility determines intestinal absorption and oral bioavailability. logS is the logarithm of the aqueous solubility value. Low solubility is detrimental to good and complete oral absorption. The value of log S of RUS was below the lower limit (log S = -5.318, Table 2 and Fig 1), suggesting that in the process of RUS development, it is necessary to focus on enhancing its log S value in order to conduct drug development more effectively. The properties of log P, log D7.4 and LogS are basic physicochemical properties of RUS, which are crucial for its design, ensuring effective delivery and improving overall therapeutic outcomes. The flexibility of the molecule has an important influence on the ADMET properties of the drug. Therefore, in the early stage of drug discovery, the flexibility of the molecule is often used as a filtering rule to screen the large-scale compound libraries, and molecules with excessive flexibility are considered less likely to act on the traditional drug targets. nRot

**Table 1. Virtual screening results of natural medicine extracts and risperidone.**

| Natural medicine extracts | Binding energy With 5-HT2AR (Kcal/mol) | Binding energy with DRD2 (Kcal/mol) | Natural medicine extracts | Binding energy With 5-HT2AR (Kcal/mol) | Binding energy With DRD2 (Kcal/mol) |
|---|---|---|---|---|---|
| RUS | -10.5 | -8.85 | Puerarin | -8.15 | -7.35 |
| risperidone | -10.8 | -10.32 | D-synephrine | -7.61 | -5.85 |
| chenodeoxycholic acid | -9.79 | -7.07 | astragalosideIV | -4.7 | -5.80 |
| columbamine | -7.8 | -8.83 | calycosin | -7.22 | -7.53 |
| coptisine | -8.61 | -8.29 | phellodendrine | -7.75 | -7.75 |
| palmatine | -7.82 | -8.21 | Cyperolone | -6.67 | -7.45 |
| berberine | -8.46 | -8.84 | cyperotundone | -7.19 | -6.6 |
| hesperidin | -7.06 | -4.09 | magnolol | -6.97 | -7.43 |
| nobiletin | -7.50 | -6.65 | Physcion-8-O-beta-D-glucopyranoside | -4.15 | -6.05 |
| tangeretin | -7.51 | -7.76 | emodin-8-o-beta-d-glucopyranoside | -4.07 | -7.46 |
| Perillaldehyde ((S)-p-Mentha-1,8-dien-7-al) | -5.41 | -6.01 | liquiritin | -9.58 | -7.68 |
| prim-o-beta-d-glucosylcimifugin | -6.29 | -4.85 | 18α-hydroxyglycyrrhetic acid | -7.87 | -7.28 |
| emodin | -7.28 | -6.69 | Cholic Acid | -8.54 | -6.69 |
| chrysophanol | -6.84 | -7.04 | beta-asarone | -4.83 | -5.05 |
| Physcion | -6.86 | -7.03 | Platycodin D | -0.02 | 2.12 |
| aloe-emodin | -6.21 | -6.98 | bilirubin | -7.26 | -7.62 |
| rhein | -6.89 | -5.86 | Muscone | -7.88 | -7.41 |
| (-)-alpha-Pinene | -6.26 | -6.59 | Harpagide | -4.11 | -3.72 |
| Notopterol | -8.27 | -7.02 | Harpagoside | -6.81 | -5.44 |
| Isoimperatorin | -7.93 | -7.26 | calceolarioside B | -4.72 | -4.74 |
| 5-O-Methylvisamminol | -7.16 | -6.8 | Baicalin | -5.85 | -5.0 |
| hesperidin | -4.76 | -5.53 | Paeoniflorin | -5.69 | -6.52 |
| isoferulic acid | -6.13 | -5.06 | ()-Bornyl acetate | -6.36 | -6.68 |
| ginsenoside-Rh1 | -4.77 | -5.03 | Acetate C-8 | -4.86 | -4.9 |
| ginsenoside Rb1 | 0.98 | -0.14 | Agarotetrol | -5.88 | -5.86 |
| Ginsenoside Re | -3.85 | -1.39 | Amygdalin | -5.7 | -2.91 |
| ferulic acid | -6.08 | -4.89 | Atractylodin | -6.32 | -6.52 |
| Jujuboside A | -1.41 | -4.37 | Naringin | -4.21 | -5.95 |
| Spinosin | -2.33 | -1.92 | Neohesperidin | -3.53 | -6.02 |
| Apigenin | -7.63 | -7.83 | paeonol | -4.97 | -5.92 |
| Schisandrin A | -6.73 | -5.98 | geniposide | -4.46 | -3.5 |
| benzoylaconine | -5.13 | -4.14 | saikosaponin a | -4.83 | -5.55 |
| benzoylmesaconine | -5.43 | -4.92 | Saikosaponin D | -4.68 | -4.41 |
| benzoylhypaconine | -5.99 | -5.82 | verbascoside | -1.16 | -3.41 |
| catalpol | -2.45 | -2.83 | echinacoside | 0.68 | 2.44 |
| Rehmaglutin D | -5.07 | -5.68 | | | |

(number of rotatable bonds) is the most common flexibility descriptor. The nRot value of RUS was 0 (Fig 1 and Table 2), indicating better rigidity and improved interaction with the drug target. nHA (H-bond acceptors) is the number of hydrogen bond acceptors. nHD (H-bond donors) is the number of hydrogen bond donors. nRing (number of Rings) is the number of rings. MaxRing (maximum Ring size) is the number of atoms in the largest ring. nHet is the number of heteroatoms. fChar (fraction of Charges) is formal charge, and nRig is the number

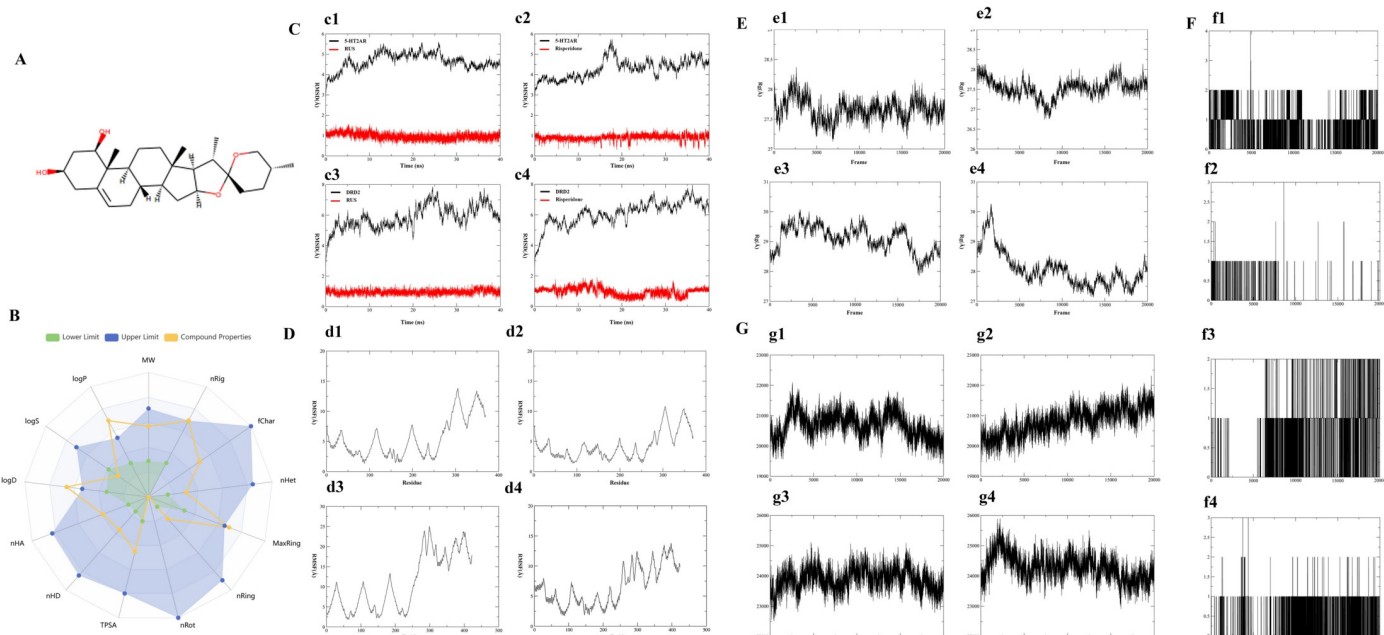

**Fig 1. Analysis of ADMET and MD simulation of RUS.** (A) The molecular structure of RUS. (B) ADMET results of RUS. (C) RMSD values. (c1) RMSD values for 5-HT2AR-RUS; (c2) RMSD values for 5-HT2AR-risperidone; (c3) RMSD values for DRD2-RUS; (c4) RMSD values for DRD2-risperidone. The red curve represents the RMSD value of the ligand, and the black curve represents the RMSD value of the protein. (D) RMSF value distribution. (d1) RMSF value distribution for 5-HT2AR-RUS; (d2) RMSF value distribution for 5-HT2AR-risperidone; (d3) RMSF value distribution for DRD2-RUS; (d4) RMSF value distribution for DRD2-risperidone. (E) Rg value distribution. (e1) Rg value distribution for 5-HT2AR-RUS; (e2) Rg value distribution for 5-HT2AR-risperidone; (e3) Rg value distribution for DRD2-RUS; (e4) Rg value distribution for DRD2-risperidone. (F) Hydrogen bonds distribution. (f1) Hydrogen bonds distribution for 5-HT2AR-RUS; (f2) Hydrogen bonds distribution for 5-HT2AR-risperidone; (f3) Hydrogen bonds distribution for DRD2-RUS; (f4) Hydrogen bonds distribution for DRD2-risperidone. (G) SASA value distribution. (g1) SASA value distribution for 5-HT2AR-RUS; (g2) SASA value distribution for 5-HT2AR-risperidone; (g3) 5-HT2AR-risperidone for DRD2-RUS; (g4) SASA value distribution for DRD2-risperidone.

of rigid bonds. TPSA (topological polar surface area) refersr to the topological polar surface area. The values for nHA (nHA = 4.0), nHD, nRing (nRing = 1.0), nHet (nHet = 4.0), fChar (fChar = 0.0), nRig (nRig = 1.0), and TPSA (TPSA = 58.92), which are the physicochemical property of RUS, fell within the optimal range (Fig 1B), while MaxRing (MaxRing = 20.0) was near the optimal range (Fig 1B). Pgp-inhibitor (P-glycoprotein Inhibitor) and HIA (human intestinal absorption) are absorption properties of RUS. The values of Pgp-inhibitor (Pgp-inhibitor = 0.024) and HIA (HIA = 0.0) indicate that RUS has good bioavailability and is well absorbed in the intestine. The blood-bain barrier is a distribution property of RUS, which shows good penetration with a value of 0.51. Metabolism properties, including CYP1A2 inhibitor (0.0), CYP1A2 substrate (0.0), CYP2C19 inhibitor (0.0), CYP2C19 substrate (0.007), CYP2C9 inhibitor (0.008), CYP2C9 substrate (0.0), CYP2D6 inhibitor (0.001), and CYP2D6 substrate (0.0) of RUS, showed excellent value, indicating that RUS performs well in its interaction with the cytochrome P450 enzyme system. In toxicity prediction, such as AMES toxicity (with a value of 0.18), eye corrosion (with a value of 0.009), hERG blockers (with a value of 0.166), respiratory toxicity (with a value of 0.25), drug-induced neurotoxicity (with a value of 0.08) and acute toxicity rule (0 rule), RUS also shows good safety. The ADMET results are shown in Fig 1B and Table 2.

## Dynamics stability assessment by RMSD

Dynamics stability influences the efficiency of biological reactions, and any perturbation in dynamics stability affects the overall reaction rate. Moreover, it also ensures the specificity of

**Table 2. ADMET analysis of RUS.**

| Property | Value | Property | Value | Property | Value |
|---|---|---|---|---|---|
| logP | 5.044 | logD7.4 | 4.272 | logS | -5.318 |
| nRot | 0.0 | nHA | 4.0 | nHD | 2.0 |
| nRing | 1.0 | MaxRing | 20.0 | nHet | 4.0 |
| fChar | 0.0 | nRig | 1.0 | TPSA | 58.92 |
| Pgp-inhibitor | 0.024 | HIA | 0.0 | Blood-bain barrier penetration | 0.51 |
| CYP1A2 inhibitor | 0.0 | CYP1A2 substrate | 0.0 | CYP2C19 inhibitor | 0.0 |
| CYP2C19 substrate | 0.007 | CYP2C9 inhibitor | 0.008 | CYP2C9 substrate | 0.0 |
| CYP2D6 inhibitor | 0.001 | CYP2D6 substrate | 0.0 | AMES toxicity | 0.18 |
| Eye corrosion | 0.009 | hERG blockers | 0.166 | Respiratory toxicants | 0.25 |
| Drug-induced neurotoxicity | 0.08 | Acute toxicity rule | 0 | | |

Note: nRot: Optimal: 0 ~ 11. nHA: Optimal: 0 ~ 12. nHD: Optimal: 0 ~ 7. nRing: Optimal: 0 ~ 6. MaxRing: Optimal: 0 ~ 18. nHet: Optimal: 1 ~ 15. fChar: Optimal: -4 ~ 4. nRig: Optimal: 0 ~ 6. TPSA: Optimal: 0 ~ 140. Pgp-inhibitor: Category 1: Inhibitor, Category 0: Non-inhibitor; the output value is the probability of being a Pgp-inhibitor. HIA: Category 1: HIA+ (HIA < 30%), Category 0: HIA- (HIA > = 30%); the output value is the probability of being HIA+. Blood-brain barrier: Category 1: BBB+, Category 0: BBB-; the output value is the probability of being BBB+. CYP1A2 inhibitor: Category 1: Inhibitor, Category 0: Non-inhibitor; the output value is the probability of being an inhibitor. CYP1A2 substrate: Category 1: Substrate, Category 0: Non-substrate; the output value is the probability of being a substrate. CYP2C19 inhibitor: Category 1: Inhibitor, Category 0: Non-inhibitor; the output value is the probability of being an inhibitor. CYP2C19 substrate: Category 1: Substrate, Category 0: Non-substrate; the output value is the probability of being a substrate. CYP2C9 inhibitor: Category 1: Inhibitor, Category 0: Non-inhibitor; the output value is the probability of being an inhibitor. CYP2C9 substrate: Category 1: Substrate, Category 0: Non-substrate; the output value is the probability of being a substrate. CYP2D6 inhibitor: Category 1: Inhibitor, Category 0: Non-inhibitor; the output value is the probability of being an inhibitor. CYP2D6 substrate: Category 1: Substrate, Category 0: Non-substrate; the output value is the probability of being a substrate. AMES toxicity: Category 1: Ames positive (+), Category 0: Ames negative (-); the output value is the probability of being toxic. Eye corrosion: Category 1: Corrosives, Category 0: Non-corrosives; the output value is the probability of being corrosive. hERG blockers: The output value is the probability of being hERG+, within the range of 0 to 1. Respiratory toxicants: The output value is the probability of being toxic, within the range of 0 to 1. Drug-induced neurotoxicity: The output value is the probability of being neurotoxic (+), within the range of 0 to 1. Acute toxicity rule: Acute toxicity during oral administration.

these interactions, effective signal transduction within the cell [40]. To assess the stability of the simulation, the RMSD of the system was calculated using the CPPTRAJ module. The results showed that there were basically no significant overall changes in the RMSD curve during the simulated 40 ns in the four systems (5-HT2AR-RUS, 5-HT2AR-risperidone, DRD2-RUS, DRD2-risperidone), indicating that those systems were stable (Fig 1C). Additionally, RUS exhibited more stable RMSD plots than risperidone (Fig 1C). To further confirm the stability of RUS binding to 5-HT2AR, we used chenodeoxycholic acid as a control natural product extract, which is the natural product extract with the highest binding affinity to 5-HT2AR identified through AutoDock Vina screening, aside from RUS. After conducting the same MD analysis as with RUS, the RMSD for the interaction between chenodeoxycholic acid and 5-HT2AR is shown in supporting information S2 File. The results indicate that the RMSD fluctuation range of chenodeoxycholic acid is greater than that of RUS and risperidone, while RUS demonstrates a more stable RMSD plot in comparison.

## Residue's flexibility calculation through RMSF

In MD simulations, the RMSF assists in pinpointing significant flexible areas pivotal in protein-ligand interactions [40]. Proteins with higher flexibility may allow ligands to enter and bind to the binding site more easily. However, the same flexibility may also allow ligands to move within or escape from the binding site, which could reduce the stability of the complex [41]. The simulated RMSF values of 5-HT2A with RUS and risperidone showed lower RMSF values for residues 50 to 260, indicating that these regions have better rigidity, which is good

for the stability of the complex (Fig 1D d1, d2). The simulated RMSF values of DRD2 with RUS and risperidone showed lower RMSF values for residues 50 to 250, indicating that these regions have better rigidity which is good for the stability of the complex (Fig 1D d3,d4).

## Protein structure compactness measurement through Rg

The Rg serves as an indicator for the plastic potential of the protein structure, and the calculation of the Rg aims to uncover the alteration in the compactness of the protein within the system throughout the simulation process, which is directly associated with its tertiary structure. The high compactness may enhance the structural stability of protein-ligand complexes, making the binding between proteins and ligands more precise. After the 40 ns MD simulation, the Rg plot value of 5-HT2AR-RUS was lower than that of 5-HT2AR-risperidone (Fig 1E e1, e2) and the Rg plot value of the DRD2-risperidone was lower than DRD2-RUS (Fig 1E e3,e4). This indicates a higher compactness of both 5-HT2AR-RUS and DRD2-risperidone, suggesting that the binding of RUS to 5-HT2AR is tighter, with a more stable and precise structure. In contrast, the binding of risperidone to DRD2 is also characterized by precision and stability compared to RUS.

## Binding strength of interactions analysis by hydrogen bonding

Hydrogen bonds were used to determine the binding strength of interactions. We estimated the hydrogen bonds in each trajectory over time. It was found that during the 40 ns simulation of 5-HT2AR and RUS, the hydrogen bonds increased, indicating that the interaction between 5-HT2AR and RUS was enhanced (Fig 1F f1). During the binding process of 5-HT2AR with risperidone, the hydrogen bond was not the main driving force (Fig 1F f2). During the 40 ns simulation of DRD2-RUS and DRD2-risperidone, the hydrogen bonds also increased, indicating that the interaction between DRD2 with RUS (Fig 1F f3) and risperidone (Fig 1F f4) also increased as the simulation proceeded.

## Protein surface feature analysis by SASA

The SASA was employed to predict the amount of residues on the surface of the protein as well as the number of residues buried within the hydrophobic core. In this study, since RUS is hydrophobic, a smaller SASA value indicates that RUS binds more tightly to the active sites of 5-HT2AR and DRD2. The SASA value of RUS with 5-HT2AR was smaller than that of risperidone, indicating that the binding of RUS to 5-HT2AR is tighter than that of risperidone to 5-HT2AR (Fig 1G g1, g2). The SASA value for RUS binding to DRD2 was consistent with that for risperidone binding to DRD2, indicating that the binding strengths of RUS and risperidone to DRD2 are similar (Fig 1G g3, g4).

## Binding free energy through MM-GBSA analysis

Through MD trajectory analysis, the binding free energy of 5-HT2AR-RUS, 5-HT2AR-risperidone, DRD2-RUS, and DRD2-risperidone complexes was estimated. The MM-GBSA analysis results showed that the binding affinity of 5-HT2AR with RUS was -43.81 kcal/mol (Fig 2A a2, Table 3), which is lower than that of risperidone, whose binding affinity was -35.68 kcal/mol (Fig 2B b2, Table 3), indicating that RUS is a more potential 5-HT2AR inhibitor than risperidone. Protein residues play a crucial role in the binding energy of protein-ligand interactions. The hydrogen bonding effect of protein residues has a significant impact on their rigidity and flexibility. Hydrogen bonds can enhance the rigidity of the protein structure [42]. Besides, the increase in the Euclidean distance or covalent bond distance between atom pairs will lead to a

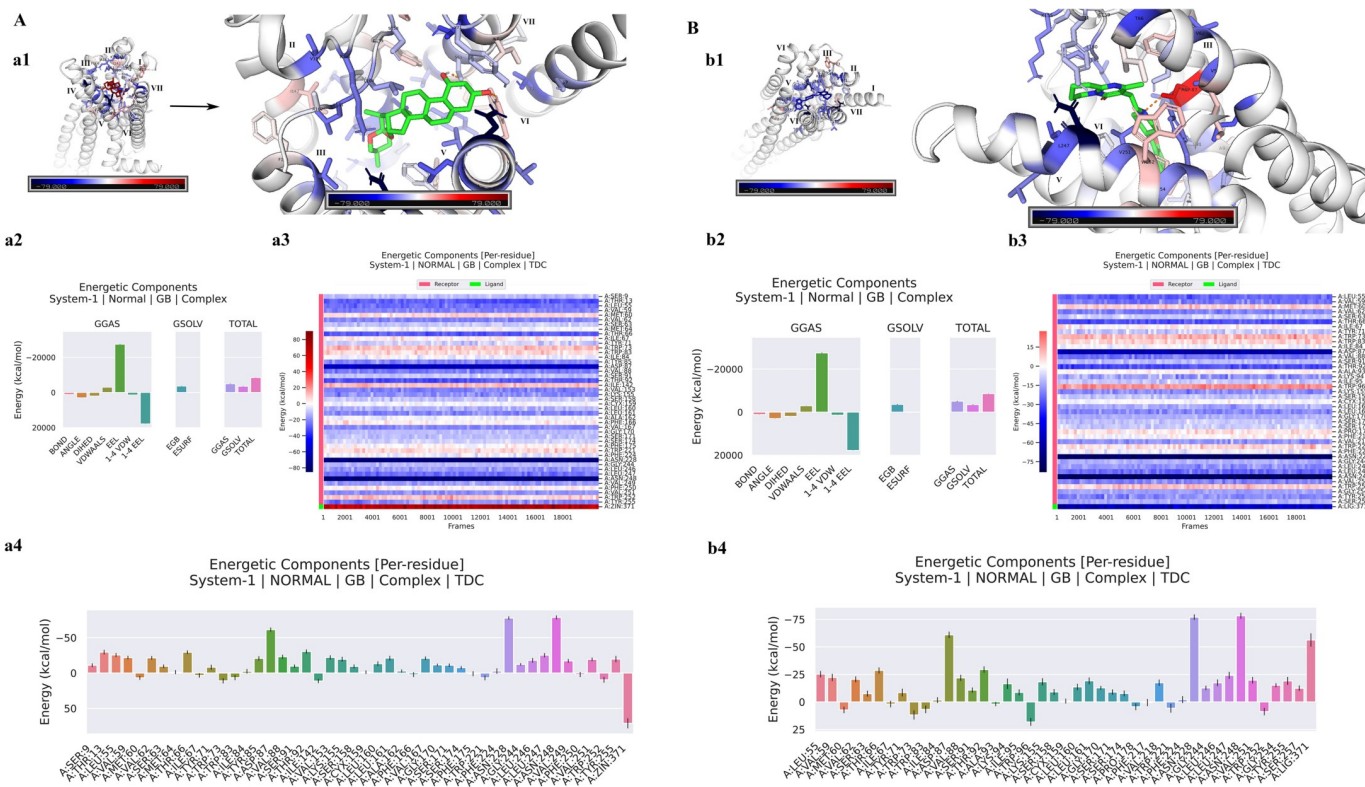

**Fig 2. The MM-GBSA analysis of 5-HT2AR with RUS and risperidone.** (**A**) The MM-GBSA analysis of 5-HT2AR with RUS. (a1) The minimum binding energy structure diagram of 5-HT2AR with RUS; (a2) Binding energy of 5-HT2AR with RUS; (a3) The heat map residue decomposition results of 5-HT2AR with RUS; (a4) The columnar diagram of residue decomposition for 5-HT2AR with RUS. (**B**) The MM-GBSA analysis of 5-HT2AR with risperidone. (b1) The minimum binding energy diagram of 5-HT2AR with risperidone; (b2) Binding energy of 5-HT2AR with risperidone; (b3) The heat map residue decomposition results of 5-HT2AR with risperidone; (b4) The columnar diagram of residue decomposition for 5-HT2AR with risperidone.

decrease of their rigidity. Covalently bonded atoms exhibit rigid body behavior, while atoms that are far apart may show considerable flexibility [43]. During the process of 5-HT2AR binding with RUS, the residues SER9, THR13, LEU55, VAL59, MET60, THR66, ILE67, TYR71, TRP73, TRP83, ILE84, TYR85, ASP87, VAL88, SER91, THR92, ILE142, VAL153, LYS155, SER158, CYX159, LEU160, LEU161, ALA162, PHE166, VAL167, GLY170, SER171, SER174, PHE175, TRP221, PHE224, ASN228, GLY244, LEU246, LEU247, ASN248, VAL249, PHE250, VAL251, TRP252, TYR255 were involved in the RUS interaction. Among them, the residues ASN248, ASN228, and ASP87 played the major role (Fig 2A a3, a4). They enhance the rigidity of specific regions of the protein, contributing to the maintenance of the stability of the structure and the accuracy of the functional sites. During the process of 5-HT2AR binding with risperidone, the residues LEU55, VAL59, MET60, VAL62, THR66, ILE67, TYR71, TRP73, TRP83, ILE84, ASP87, VAL88, SER91, THR92, ALA93, LYS94, ILE95, TRP96, LYS155, SER158, CYX159, LEU160, LEU161, GLY170, SER171, SER174, PRO178, PHE217, VAL218, TRP221, PHE224, ASN228, GLY244, LEU246, LEU247, ASN248, VAL251, TRP252, TYR255, and SER257 were involved in the RUS interaction. Among them, the residues ASN248, ASN228, and ASP87 also played a major role (Fig 2B b3, b4). The outcomes indicate that the mechanism by which RUS and risperidone operate on 5-HT2AR is identical; yet RUS, as a 5-HT2AR inhibitor, possesses a superior effect. The minimum binding energy structure diagram of 5-HT2AR with RUS is shown in Fig 2A a1, and the minimum binding energy diagram of 5-HT2AR with risperidone is shown in Fig 2B b1.

**Table 3. Binding energy of 5-HT2AR with RUS and risperidone.**

| Complex | ΔVDWAALS (Kcal/mol) | ΔEEL (Kcal/mol) | ΔEGB (Kcal/mol) | ΔESURF (Kcal/mol) | ΔGGAS (Kcal/mol) | ΔGSOLV (Kcal/mol) | ΔTOTAL (Kcal/mol) |
|---|---|---|---|---|---|---|---|
| 5-HT2AR-RUS | -50.61 | -10.18 | 21.15 | -4.17 | -60.79 | 16.98 | -43.81 |
| 5-HT2AR-risperidone | -49.48 | -9.28 | 27.38 | -4.29 | -58.77 | 23.08 | -35.68 |

Note: ΔVDWAALS: van der waals energy. ΔEEL: electrostatic energies. ΔEGB: polar solvation energy. ΔESURF: Non-polar solvation energy. ΔGGAS: ΔVDWAALS +ΔEEL. ΔGSOLV = ΔEGB+ΔESURF. ΔTOTAL = ΔGSOLV+ΔGGAS.

The relative binding free energies for DRD2 with RUS and risperidone were -35.00 kcal/mol (Fig 3C c2, Table 4) and -46.05 kcal/mol (Fig 3D d2, Table 4), respectively. The residue decomposition results showed that during the binding process of DRD2 with RUS, the residues TYR3, LEU7, TRP56, VAL57, VAL58, TYR59, LEU60, GLU61, VAL62, VAL63, GLY64, GLU65, TRP66, PHE76, VAL77, ASP80, VAL81, ILE146, VAL152, SER155, TRP207, PHE210, THR213, HIE214, ASN217, PRO226, VAL227, LEU228, TYR229, SER230, PHE232, THR233, TRP234, and TYR237 in DRD2 participated in the interaction with RUS. Among them, residues GLU61, GLU65, ASP80, and ASN217 played a major role (Fig 3C c3, c4). They enhance the rigidity of specific regions of the protein, contributing to the maintenance of the stability

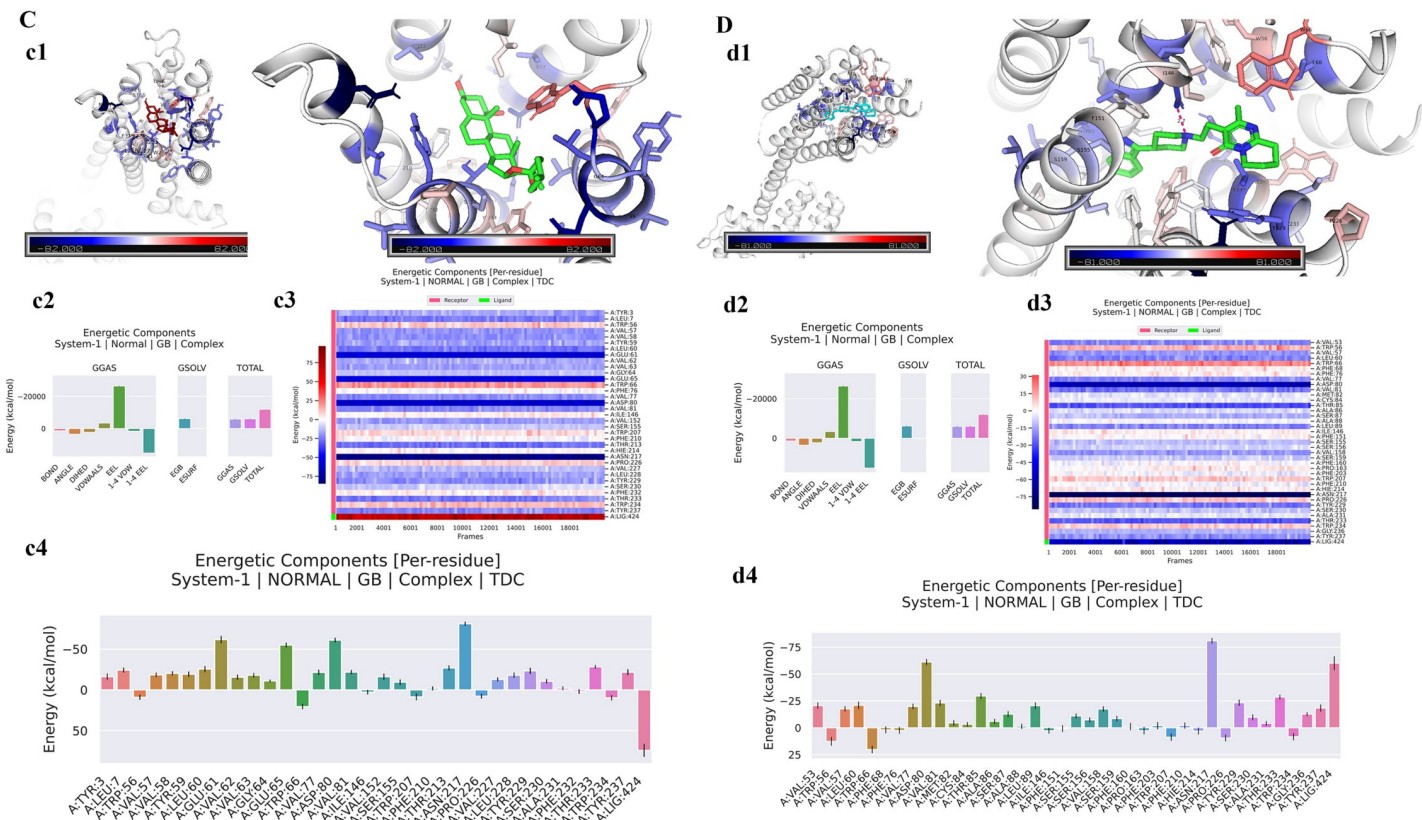

**Fig 3. The MM-GBSA analysis of DRD2 with RUS and risperidone.** (A) The MM-GBSA analysis of DRD2 with RUS. (a1) The minimum binding energy structure diagram of DRD2 with RUS. (a2) Binding energy of DRD2 with RUS. (a3) The heat map residue decomposition results of DRD2 with RUS; (a4) The columnar diagram of residue decomposition for DRD2 with RUS; (B) The MM-GBSA analysis of DRD2 with risperidone. (b1) The minimum binding energy diagram of DRD2 with risperidone. (b2) Binding energy of DRD2 with risperidone. (b3) The heat map residue decomposition results of DRD2 with risperidone; (b4) The columnar diagram of residue decomposition for DRD2 with risperidone.

**Table 4. Binding energy of DRD2 with RUS and risperidone.**

| Complex | ΔVDWAALS (Kcal/mol) | ΔEEL (Kcal/mol) | ΔEGB (Kcal/mol) | ΔESURF (Kcal/mol) | ΔGGAS (Kcal/mol) | ΔGSOLV (Kcal/mol) | ΔTOTAL (Kcal/mol) |
|---|---|---|---|---|---|---|---|
| DRD2-RUS | -44.25 | -19.87 | 32.97 | -3.84 | -64.13 | 29.12 | -35.00 |
| DRD2-risperidone | -54.68 | -11.02 | 24.19 | -4.54 | -65.70 | 19.65 | -46.05 |

Note: ΔVDWAALS: van der waals energy. ΔEEL: electrostatic energies. ΔEGB: polar solvation energy. ΔESURF: Non-polar solvation energy. ΔGGAS: ΔVDWAALS +ΔEEL. ΔGSOLV = ΔEGB+ΔESURF. ΔTOTAL = ΔGSOLV+ΔGGAS.

of the structure and the accuracy of the functional sites. During the binding process of DRD2 with risperidone, the residues VAL53, TRP56, VAL57, LEU60, TRP66, PHE68, PHE76, VAL77, ASP80, VAL81, MET82, CYS84, THR85, ALA86, SER87, ALA88, LEU89, ILE146, PHE151, SER155, SER156, VAL158, SER159, PHE160, PRO163, PHE203, TRP207, PHE210, HIE214, ASN217, PRO226, TYR229, SER230, ALA231, THR233, TRP234, GLY236, and TYR237 in DRD2 participated in the interaction with risperidone. Among them, residues ASP80 and ASN217 residues played a major role (Fig 3D d3, d4). The results indicated that the mechanisms by which RUS and risperidone act on DRD2 are similar, and residues ASP80 and ASN217 are both the main residues that play the major role. Although RUS does not have a higher binding energy with DRD2 than risperidone, it also plays a role in inhibiting DRD2. The minimum binding energy structure diagram of DRD2 with RUS is shown in Fig 3C c1, and the minimum binding energy diagram of DRD2 with risperidone is shown in Fig 3D d1.

### Protein local structure and dynamics analysis by DSSP

DSSP provides an understanding of the local structure and dynamics of biomolecules, involving the identification of sections of the biomolecule that exhibit specific types of local structure, such as alpha helices, beta sheets, and turns, which affect its function and interactions with other biomolecules [44]. In the 40 ns simulation study, 5-HT2AR and DRD2 conserved their overall secondary structure, with minimal deviation from their initial positions (Fig 4). During the binding process of 5-HT2AR with RUS, the structure of residue VAL251 mainly fluctuated between turn and alpha, while the residues SER174 and SER171 formed alpha helices, and LEU160 formed a bond structure. ILE67 changed from an unstable turn structure to stable turn structure, and THR66 and MET64 changed from unstable alpha helices to stable alpha helices (Fig 4A). During the binding process of 5-HT2AR with risperidone, the residues VAL251 and ASN248 generated alpha helices, while the alpha helices of THR64 and SER63 gradually disappeared (Fig 4B). During the binding process of DRD2 with RUS, the turn structure of residue TRP234 and the alpha helices of THR233 gradually disappeared as the simulation proceed. The alpha helices of the residue PHE232 transformed into bond and turn structures, the alpha helices of SER230 and TYR229 transformed into turn structures, the residue GLY64 changed between bond and turn structures (Fig 4C). During the binding process of DRD2 with risperidone, the residues SER159, VAL158, and SER156 changed from unstable turn structures to stable turn structures (Fig 4D).

### Discussion

Pharmacotherapy is essential for managing the symptoms of psychotic disorders, with a variety of antipsychotic drugs developed over the past seven decades. The DRD2 is a primary target for pharmacogenetic research on antipsychotic drugs. While DRD2 antagonists can effectively alleviate positive symptoms, they are often ineffective against treatment-resistant cases and

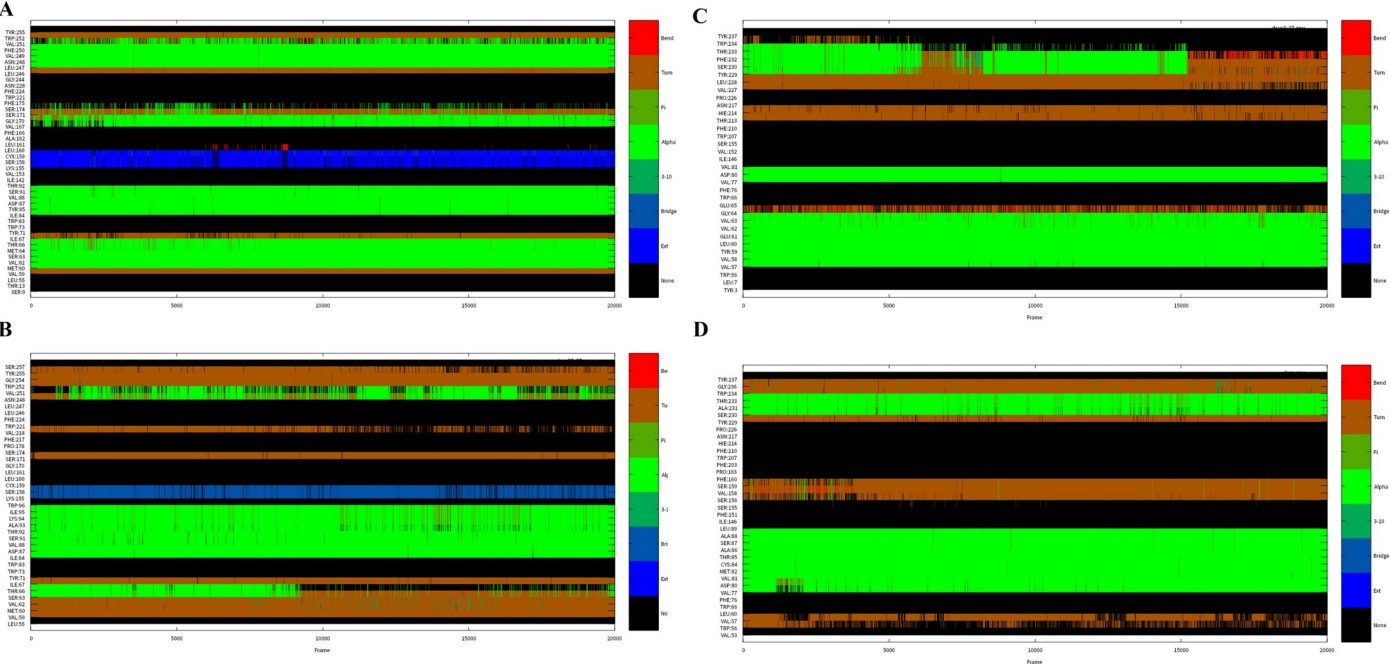

**Fig 4. The secondary structural analysis of 5-HT2AR and DRD2 amino acid residues during MD simulation using the DSSP method.**

have a limited impact on negative symptoms. Additionally, these drugs may pose risks of significant neurological side effects [45]. Newer 'atypical' antipsychotic drugs, which act as antagonists of both the DRD2 and 5-HT2AR, have demonstrated a decreased likelihood of neurological adverse effects while still effectively reducing positive symptoms. However, these atypical antipsychotics are linked to notable metabolic disruptions and weight gain [46]. Furthermore, individuals with cardiovascular and cerebrovascular conditions may also experience mental health issues such as anxiety and depression, underscoring the importance of exploring safe and efficacious natural remedies or their derivatives in addressing psychotic disorders like anxiety and depression.

Our research indicates that RUS has a higher binding energy compared to other natural product extracts (the binding affinity of RUS with 5HT2AR and DRD2 was -10.5 kcal/mol and -8.85 kcal/mol, respectively; see Table 1). RUS is a major bioactive steroidal sapogenin extracted from the natural product *radix ophiopogon japonicus*. Study has shown that RUS has the ability to increase fluidity of membranes and has significant interaction with and penetration into biological membranes, which are basic conditions for therapeutic drug molecules and present the biggest challenges in drug development [47]. To further investigate the mechanism by which RUS antagonizes 5-HT and DA receptors, this study employed MD analysis to elucidate the interactions of RUS and risperidone bound to the 5-HT2AR and DRD2. MD analysis facilitated a clear understanding of the relationship between the chemical structures of RUS and risperidone and their target activities. The MD analysis revealed that RUS has potent antagonistic effects on the 5-HT and DA receptors, with the binding affinity of RUS with 5-HT2AR and DRD2 being -43.81 kcal/mol and -35.00 kcal/mol, respectively. The ADMET analysis revealed favorable drug-like properties of RUS within a significant range, indicating its suitability as a drug candidate. RUS exhibited enhanced liposolubility with high logP and logD7.4 values, indicating that RUS has good bioavailability and higher pharmacokinetic characteristics. RUS showed favorable parameters such as nHBAs and nHBDs, indicating that RUS

has good molecular interactions. RUS has a reasonable nRotb value, which can reduce the complexity of molecular conformations, helping to improve drug stability and safety while lowering toxicity risks. Additionally, assessments of Pgp-inhibitor and HIA suggest that RUS possesses good bioavailability and intestinal absorption. RUS demonstrated effective blood-brain barrier penetration (value 0.51) and interaction with cytochrome P450 enzyme system. Acute toxicity rule was not observed with RUS, providing evidence of its good safety profile. Overall, the ADMET parameters of RUS obtained from the predicted ADMET data were satisfactory, confirming RUS's ability to mimic the properties of drugs or leads.

Risperidone is a second-generation antipsychotic with selective antagonistic properties, acting against the 5-HT2AR and DRD2. Currently, risperidone is widely used in the clinical treatment of psychiatric disorders [48]. Regarding its mechanism of action on 5-HT2AR and DRD2, in the case of conformational stability the conformation stabilizes (as indicated by achieving equilibrium in RMSD across each MD trajectory), the simulated RMSF values of RUS with 5-HT2A were lower than those of risperidone, with lower RMSF values lying in residues 50 to 260. The simulated RMSF values of RUS with DRD2 were similar to those of risperidone, with lower RMSF values lying in residues 50 to 250. The Rg plot of the 5-HT2AR-RUS complex was lower than that of the 5-HT2AR-risperidone complex, which indicated that RUS has a more compact structure with 5-HT2AR than risperidone. The Rg value of the DRD2-risperidone complex was lower than that of DRD2-RUS complex, indicating that risperidone has a more compact structure with DRD2 than RUS. These findings indicate that the mechanism of action of RUS is similar to that of risperidone, which can stably bind with 5-HT2AR and DRD2 and has a more stable structure than risperidone when binding to 5-HT2AR.

The more hydrogen bonds between the complexes, the more stable they are. It was found that the hydrogen bonds increased during the simulation of RUS with 5-HT2AR and DRD2, indicating that the interaction between 5-HT2AR-RUS and DRD2-RUS were enhanced. Since RUS is hydrophobic, a smaller SASA value indicates that RUS binds more tightly to the active sites of 5-HT2AR and DRD2. The SASA values of RUS with 5-HT2AR were smaller than those of risperidone, suggesting that RUS binds more tightly to 5-HT2AR than risperidone dose to 5-HT2AR. The binding of RUS to DRD2 is consistent with the SASA values of risperidone binding to DRD2, indicating that the binding tightness of RUS and risperidone to DRD2 is similar.

MM-GBSA analysis showed that 5-HT2AR had a lower binding affinity with RUS than with risperidone, suggesting that RUS is a more promising 5-HT2AR inhibitor than risperidone. The residues ASN248, ASN228, and ASP87 in 5-HT2AR play a major role in the binding of 5-HT2AR to RUS and risperidone. These results suggested that RUS and risperidone act on 5-HT2AR by essentially the same mechanism, but RUS is more effective as a 5-HT2AR inhibitor. The relative binding free energies of RUS and risperidone to DRD2 were −35.00 kcal/mol and −46.05 kcal/mol, respectively. The residues GLU61, GLU65, ASP80, and ASN217 in DRD2 play a major role in the binding of DRD2 to RUS. The residues ASP80 and ASN217 in DRD2 play a major role in the binding of DRD2 to risperidone. The results showed that the mechanism by which RUS and risperidone acted on DRD2 was essentially the same, with ASP80 and ASN217 residues being the main residues. Although RUS does not have the high binding energy of risperidone to DRD2, it still plays a role in inhibiting DRD2.

The changes in the secondary structure of the protein in different systems mainly involve changes in the alpha helixs. Using DSSP analyzes the secondary structure of the protein, the results showed that 5-HT2AR and DRD2 conserved their overall secondary structure with minimal deviation from their initial positions. In the process of binding with RUS, the structure of the residue VAL251 in 5-HT2AR mainly fluctuates between a turn and alpha helix, while the residues SER174 and SER171 formed alpha helices, and THR66 and MET64 changed

from unstable alpha helices to stable alpha helices. During the binding with risperidone, the residue VAL251 and ASN248 in 5-HT2AR generated the alpha helices. While the alpha helices of THR64 and SER63 gradually disappeared. In the process of binding with RUS, the turn structure of the residue TRP234 and the alpha helices of THR233 in DRD2 gradually disappeared as the simulation progressed. The alpha helices of the residue PHE232 transformed into bond and turn structures, and the alpha helices of SER230 and TYR229 transformed into the turn structure. In the binding process of DRD2 with risperidone, the residues SER159, VAL158, and SER156 changed from unstable turn structures to stable turn structures.

In recent years, natural products and their extracts have been found to have potential applications in treating psychiatric behavioral symptoms [49]. The methanol extract of *Magnolia officinalis*, containing magnolol and honokiol, has been identified as antagonists of 5-HT2AR and DRD2 [50]. Additionally, 21-hydroxyshidasterone, isolated from the methanol stem bark extract of *V. doniana*, exhibits antidepressant-like effects through interactions with 5-HT2AR and DRD2 [51]. The degradation products of quercetin may exert antidepressant effects via the serotonergic system [52]. Studies using molecular docking have shown that myristic acid can form favorable interactions within the binding pockets of the 5-HT2A receptor [53]. Research has further indicated that bioactive compounds such as ginsenosides, quercetin, curcumin, and rutin can bind to DRD2 protein with confirmed binding affinity scores [54]. Among these compounds, RUS and ginsenosides are both classified as saponins; RUS is a steroid saponin, while ginsenosides belong to triterpene saponins. RUS has shown various biological activities including anti-inflammation [55], anti-tumor [56], anti-thrombosis formation [57], cardiac protection [58] and blood-brain barrier dysfunction [59], all without reported side effects. In this study, RUS antagonizes 5-HT2AR and DRD2, which may exert therapeutic effects on psychiatric behavioral symptoms. Compared to risperidone, a second-generation antipsychotic, RUS may have broader applications; it may be used to treat tumors accompanied by psychiatric behavioral symptoms, as well as anxiety and depression associated with cardiovascular diseases and psychiatric behavioral symptoms resulting from blood-brain barrier dysfunction. Animal studies have reported that RUS can improve anxiety-like behavior in complete Freund's adjuvant-induced mice [16], showing anxiety-relieving effects comparable to the positive control drug clonazepam, and it also possesses anti-neuroinflammatory properties. A study conducted an 3-(4,5)-dimethylthiahiazo (-z-y1)-3,5-di- phenytetrazoliumromide (MTT) assay, and the results indicated that RUS does not produce significant cytotoxicity on mouse microglial cells, even at a higher concentration of 100 μM/ml [60]. In this study, MD simulations revealed the specific mechanisms of RUS binding to 5-HT2AR and DRD2, including binding strength and antagonistic pattern side effects. Although MD offers important theoretical foundations, clinical trials are still needed to validate the practical efficacy of these findings. Prospective studies are recommended to assess the effectiveness and safety of RUS in real-world settings.

## Conclusions

Our study offers a potential natural extract for treating psychiatric behavioral symptoms, which has demonstrated therapeutic effects in various diseases. The results indicate that RUS exhibits superior antagonistic effects on 5-HT2AR and DRD2 compared to other natural product extracts. Preclinical predictions suggest that RUS possesses favorable drug-like properties, and its mechanism of action in antagonizing 5-HT2AR and DRD2 is consistent with that of risperidone, with even better efficacy against 5-HT2AR. Animal experiments have confirmed that RUS can improve anxiety-like behavior in complete Freund's adjuvant-induced mice [16], showing anxiety-relieving effects comparable to those of the positive control drug clonazepam.

This further emphasizes the potential therapeutic role of RUS in addressing psychiatric disorders and other conditions associated with abnormal dopamine and serotonin levels. RUS is a promising candidate for treating psychiatric diseases and the accompanying behavioral symptoms. Furthermore, subsequent experiments and clinical studies to assess its safety and efficacy, as well as to bring this potential therapy to market, are both effective and necessary.

## Supporting information

**S1 Fig. 2D structure of natural product extracts.**
(PDF)

**S1 File. S2-29 Figs.** Molecular docking results of natural product extracts with 5-HT2AR and DRD2.
(PDF)

**S2 File. S30 Fig.** RMSD values for 5-HT2AR-chenodeoxycholic acid.
(PDF)

## Acknowledgments

The work was carried out at National Supercomputer Center in Tianjin, and the calculations were performed on Tianhe new generation supercomputer.

## Author Contributions

**Conceptualization:** Yongmei Liu.

**Investigation:** Yongmei Liu.

**Writing – original draft:** Suya Ma.

**Writing – review & editing:** Yongmei Liu.

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
