## [Decision Letter · Decision Letter 0]

15 Aug 2024

PONE-D-24-30011Natural extracts in psychiatry: Ruscogenin's role in antagonizing 5-HT2A and DA2 receptors through computational screeningPLOS ONE

Dear Dr Liu,

Thank you for submitting your manuscript to PLOS ONE. After careful consideration, we feel that it has merit but does not fully meet PLOS ONE’s publication criteria as it currently stands. Therefore, we invite you to submit a revised version of the manuscript that addresses the points raised during the review process.

We look forward to receiving your revised manuscript.

Kind regards,

Akingbolabo Daniel Ogunlakin, Phd

Academic Editor

PLOS ONE

Journal Requirements:

2. Please note that PLOS ONE has specific guidelines on code sharing for submissions in which author-generated code underpins the findings in the manuscript. In these cases, all author-generated code must be made available without restrictions upon publication of the work. 

Please review our guidelines at https://journals.plos.org/plosone/s/materials-and-software-sharing#loc-sharing-code and ensure that your code is shared in a way that follows best practice and facilitates reproducibility and reuse.

"This research was funded by National Natural Science Foundation of China (Key Program) (NO. 82230124)."

5. Please note that funding information should not appear in the Acknowledgments section or other areas of your manuscript. We will only publish funding information present in the Funding Statement section of the online submission form. Please remove any funding-related text from the manuscript. 

7. PLOS requires an ORCID iD for the corresponding author in Editorial Manager on papers submitted after December 6th, 2016. Please ensure that you have an ORCID iD and that it is validated in Editorial Manager. To do this, go to ‘Update my Information’ (in the upper left-hand corner of the main menu), and click on the Fetch/Validate link next to the ORCID field. This will take you to the ORCID site and allow you to create a new iD or authenticate a pre-existing iD in Editorial Manager. Please see the following video for instructions on linking an ORCID iD to your Editorial Manager account: https://www.youtube.com/watch?v=_xcclfuvtxQ

**Additional Editor Comments:**

The study was designed to investigate the mechanism of action of Ruscogenin in inhibiting 5 - HT2A receptor and DA2 receptor. The result highlighted the mechanisms using computational approaches with appropriate data supporting the conclusion.

However, the following comments should be looked into:

The entire article should be checked for grammatical errors and appropriate use of punctuation

Write all abbreviations in full at first mention

The aim should be stated in the abstract

Line 36-41 should be the last sentence in the introduction section, because that is the aim of the study.

What informed the use of Ruscogenin? That should be clearly stated in the introduction

Ruscogenin has been shown to possess the ability………

A sentence should not begin with abbreviations, check lines 42, 45, 46, 48, 51. 53, 62

Line 43-44: Abnormalities in the functioning of the DA system can lead to different diseases such as ………..

Line 46-47: DRD2, the second most abundant dopaminergic receptor, is mainly expressed………..

Lines 48-49: DRD2 which is critically involved …………..

Line 51-52: DRD2 is a taget for drugs used in the treatment of numerous pathologies

Line 57: Brain 5-HT2AR shows…….

The sentences in line 68-70 should be rephrased

Line 97 should be rephrased using reported speech.

Appropriate tenses should be used in the manuscript, especially the result section.

Line 189: Protein structure size measurement through Rg

What is Rg? It should be written in full before abbreviating.

Line 209: SASA should be written in full

Line 253: DSSP should be written in full

Line 309: Reference should be added

Line 352-354: In conclusion, the RUS, a safe and effective natural extract, can antagonize 5 - HT2AR and DRD2 receptors, and plays a potential therapeutic role in addressing psychiatric disorders and other conditions associated with abnormal DA and 5- HT levels

Reviewers' comments:

Reviewer's Responses to Questions

**Comments to the Author**

1. Is the manuscript technically sound, and do the data support the conclusions?

Reviewer #1: Yes

Reviewer #2: Yes

2. Has the statistical analysis been performed appropriately and rigorously? 

Reviewer #1: N/A

Reviewer #2: N/A

3. Have the authors made all data underlying the findings in their manuscript fully available?

Reviewer #1: Yes

Reviewer #2: Yes

4. Is the manuscript presented in an intelligible fashion and written in standard English?

Reviewer #1: Yes

Reviewer #2: Yes

5. Review Comments to the Author

Reviewer #1: Review of

“Natural extracts in psychiatry: Ruscogenin's role in antagonizing 5-HT2A and DA2 receptors through computational screening”

The search for effective and safe treatments for psychiatric disorders is an ongoing challenge in the field of medicine. Traditional antipsychotic drugs, although beneficial in managing symptoms, are often associated with significant adverse effects, including neurological complications, weight gain, and metabolic disruptions. These limitations underscore the urgent need for new therapeutic strategies that can offer improved efficacy and safety profiles. Natural products have long been recognized for their potential as sources of novel pharmaceuticals due to their diverse chemical structures and biological activities. Among these, ruscogenin (RUS), a steroidal sapogenin extracted from Radix Ophiopogon japonicus, has emerged as a candidate of interest. In this context, the submitted study aims to evaluate the potential of RUS as an effective inhibitor of the 5-HT2A and DA2 receptors, both of which are crucial in the pathology of psychiatric disorders such as schizophrenia and depression. Utilizing a combination of virtual molecular docking, ADMET analysis, and molecular dynamics (MD) simulations, the research seeks to explore the binding characteristics and drug-like properties of RUS compared to an established antipsychotic, risperidone.

While the study is promising and could potentially make a significant impact, several areas require revision to enhance the overall quality and clarity of the manuscript. Specifically, the following comments should be addressed:

Title:

- The title is long and contains multiple complex terms, which might make it difficult for readers to quickly grasp the main focus of the study.

- The title does not specify what RUS is, which can be confusing for readers unfamiliar with the term.

- Terms like "5-HT2AR" and "DRD2" are highly specific and may not be immediately understood: Technical Jargon

Abstract:

- Abstracts should summarize the key findings without overwhelming the reader with technical specifics.

- Some sentences are lengthy and complex, which can make the abstract harder to read.

- While the abstract mentions that RUS showed favourable properties and stability, it does not quantify these results or provide specific data points, which could help in understanding the magnitude of the findings.

- The conclusion suggests that RUS is a promising candidate for therapeutic development based on the in-silico results. However, it would be prudent to temper this with a statement about the need for further validation through in vitro and in vivo studies.

Introduction:

- The introduction covers a wide range of topics, from the general benefits of natural products to specific details about ruscogenin (RUS) and its potential therapeutic effects. While comprehensive, the flow can be confusing. A clearer structure with more defined sections would improve readability.

- The introduction is quite long and contains a lot of detailed information. While detail is important, some of the more specific data (chemical formulas, specific biological activities) might be better suited for later sections.

- Some sections, such as the detailed description of dopamine (DA) and serotonin (5-HT) receptors and their related pathologies, while informative, are quite extensive. This information might be better condensed to maintain focus on the main topic.

- The introduction lacks sufficient citations to support the claims made, especially in the initial paragraphs discussing the effectiveness of natural products and the biological activities of RUS.

- Transitions between different sections could be smoother.

- The introduction does not clearly state the objective or hypothesis of the study until the very end. Introducing the aim of the research earlier would provide context for the detailed information that follows.

Methodology

- The specific version of Autodock Vina should be clarified. If it is 1.2.0, ensure the correct reference is provided.

- Details on the parameters used for hydrogenation, charge calculation, and protonation should be more specific.

- The method for defining the binding pocket should be more explicitly described. Are specific residues or a particular region targeted?

- Explanation needed on why semi-flexible docking was chosen and how flexibility was introduced.

- Justification for selecting leaprc.protein.ff99SB and leaprc.gaff force fields. Are there specific advantages for this study?

- The rationale behind choosing a 12 Å gap and the TIP3P water model should be provided.

- Specify the concentration of Na+ or Cl- ions added.

- Explain why SHAKE is used specifically for hydrogen atoms.

- Explain why alanine scanning was chosen and how it helps in residue decomposition analysis.

- Provide clear explanations and justifications for each step and parameter choice to enhance reproducibility and understanding.

Results:

- The section mentions that "RUS was the best inhibitor," but it lacks details on the criteria used to determine this. Specify the binding affinity values or other metrics that led to this conclusion.

- There is a good level of detail regarding liposolubility, log P, log D7.4, solubility, and flexibility. However, the relationship between these properties and their practical implications for drug development should be more explicitly connected.

- The explanation of ADMET properties could benefit from more context regarding their relevance to drug development and specific thresholds that define "good" or "poor" properties.

- The results indicate stability but lack comparative data with controls or other compounds, which would help contextualize the stability of RUS and risperidone complexes.

- The explanation of RMSF results is clear, but more detail on the functional implications of the observed flexibility or rigidity in specific residues would be helpful.

- Relate the observed flexibility to the binding efficiency or stability of the protein-ligand complexes.

- The section provides a good description of the Rg values. However, it should explain the implications of high or low compactness on the function or interaction of the protein-ligand complexes.

- The comparative analysis between RUS and risperidone could be expanded to discuss why these differences in compactness matter.

- Include more precise language when describing scientific results, e.g., "indicating that it has good liposolubility and is easy to penetrate the cellular membrane structure" should be supported by numerical values and comparative benchmarks.

- Avoid vague statements.

Discussion:

- The discussion mentions that RUS showed the best inhibitory effect on 5-HT2AR and DRD2 but does not provide specific comparative metrics or data. Clear numerical results or statistical analysis would strengthen the claim.

- While the ADMET properties of RUS are discussed in detail, the practical implications of these properties in clinical settings are not fully explored. It would be beneficial to compare these properties more directly with those of existing drugs like risperidone to highlight the advantages and potential drawbacks.

- The discussion focuses on the binding properties and ADMET analysis but does not sufficiently address the clinical efficacy of RUS compared to risperidone.

- The discussion claims that RUS has a natural advantage in terms of safety but does not provide detailed safety data or potential side effects. Any adverse effects observed in preclinical or early clinical trials should be discussed. (if available)

- The discussion touches upon metabolic disruptions and weight gain associated with atypical antipsychotics but does not address whether RUS also carries these risks. Given the importance of side effects in the treatment of psychotic disorders, this aspect needs more attention.

- There is a gap in translating the molecular and computational findings into clinical relevance. How these findings might impact treatment protocols, patient outcomes, or the development of new therapies should be discussed.

- The discussion could benefit from a broader literature context. How do the findings about RUS fit within the current research landscape? Are there similar compounds being investigated? What are the next steps in the research and development process?

Conclusion:

- The conclusion repeats several points already mentioned in the discussion without adding new insights. Streamlining the conclusion to avoid redundancy would make it more concise and impactful.

- The conclusion mentions that RUS is a "potential therapeutic" but does not provide any context on the stages of clinical testing or the next steps required to bring this potential therapy to market. Discussing the clinical relevance and future directions would add depth. (in brief)

- The statement "RUS, a safe and effective natural extract" is an overgeneralization. The study should specify that safety and efficacy are inferred from preclinical analyses and that clinical trials are necessary to confirm these properties.

- The conclusion would benefit from a clearer structure. Presenting the findings in a logical sequence (starting with the discovery, moving to the detailed results, and ending with the broader implications and future directions) would improve readability.

- The conclusion lacks a connection to existing literature.

References: The paper should be enriched with more citations.

Figures: Authors should provide figures with higher resolution

Supplementary files: the intra-complexes interactions and bonds must be studied and represented (2D interaction visualization).

Reviewer #2: The study was designed to investigate the mechanism of action of Ruscogenin in inhibiting 5 - HT2A receptor and DA2 receptor. The result highlighted the mechanisms using computational approaches with appropriate data supporting the conclusion.

However, the following comments should be looked into:

The entire article should be checked for grammatical errors and appropriate use of punctuation

Write all abbreviations in full at first mention

The aim should be stated in the abstract

Line 36-41 should be the last sentence in the introduction section, because that is the aim of the study.

What informed the use of Ruscogenin? That should be clearly stated in the introduction

Ruscogenin has been shown to possess the ability………

A sentence should not begin with abbreviations, check lines 42, 45, 46, 48, 51. 53, 62

Line 43-44: Abnormalities in the functioning of the DA system can lead to different diseases such as ………..

Line 46-47: DRD2, the second most abundant dopaminergic receptor, is mainly expressed………..

Lines 48-49: DRD2 which is critically involved …………..

Line 51-52: DRD2 is a taget for drugs used in the treatment of numerous pathologies

Line 57: Brain 5-HT2AR shows…….

The sentences in line 68-70 should be rephrased

Line 97 should be rephrased using reported speech.

Appropriate tenses should be used in the manuscript, especially the result section.

Line 189: Protein structure size measurement through Rg

What is Rg? It should be written in full before abbreviating.

Line 209: SASA should be written in full

Line 253: DSSP should be written in full

Line 309: Reference should be added

Line 352-354: In conclusion, the RUS, a safe and effective natural extract, can antagonize 5 - HT2AR and DRD2 receptors, and plays a potential therapeutic role in addressing psychiatric disorders and other conditions associated with abnormal DA and 5- HT levels

6. PLOS authors have the option to publish the peer review history of their article (what does this mean?). If published, this will include your full peer review and any attached files.

Reviewer #1: **Yes: **Amel Elbasyouni

Reviewer #2: No

---

## [Author Response · Author response to Decision Letter 0]

2 Sep 2024

Dear Editor Akingbolabo Daniel Ogunlakin and Reviewers,

Thank you very much for taking the time to review this article. I especially appreciate your assistance and support. I am grateful for all your comments and suggestions! Based on your feedback, I have made thorough revisions and addressed each point meticulously. Please find my detailed responses below, along with my revisions and corrections in the resubmitted files. Thanks again!

SUGGESTIONS FROM EDITOR

Q1: Please ensure that your manuscript meets PLOS ONE's style requirements, including those for file naming. 

The author’s answer: Thank you for pointing out the formatting issues in the paper. According to your suggestions, we have made the necessary revisions in accordance with the PLOS ONE style templates.

Q2: Please note that PLOS ONE has specific guidelines on code sharing for submissions in which author-generated code underpins the findings in the manuscript. In these cases, all author-generated code must be made available without restrictions upon publication of the work. 

The author’s answer: Thank you for your comments on the code for the paper. Our code has been uploaded to protocols.io (DOI: dx.doi.org/10.17504/protocols.io.261ge5koyg47/v1).

Q3: We note that the grant information you provided in the ‘Funding Information’ and ‘Financial Disclosure’ sections do not match. 

The author’s answer: We apologize for this error. We have removed the Funding Information from the paper as required by the fifth question. We carefully verified the grant number: The grant number is 82230124.

Q4: Thank you for stating the following financial disclosure: 

"This research was funded by National Natural Science Foundation of China (Key Program) (NO. 82230124)."

The author’s answer: We apologize for not clearly stating the role of the funders. The corresponding author, Yongmei Liu, is one of the principal investigators of the project and has received funding support from them. This information has been included in the cover letter.

Q5: Please note that funding information should not appear in the Acknowledgments section or other areas of your manuscript. We will only publish funding information present in the Funding Statement section of the online submission form. Please remove any funding-related text from the manuscript. 

The author’s answer: We apologize for including the funding information in the manuscript. We have since removed this information.

Q6: When completing the data availability statement of the submission form, you indicated that you will make your data available on acceptance. We strongly recommend all authors decide on a data sharing plan before acceptance, as the process can be lengthy and hold up publication timelines. Please note that, though access restrictions are acceptable now, your entire data will need to be made freely accessible if your manuscript is accepted for publication. This policy applies to all data except where public deposition would breach compliance with the protocol approved by your research ethics board. If you are unable to adhere to our open data policy, please kindly revise your statement to explain your reasoning and we will seek the editor's input on an exemption. Please be assured that, once you have provided your new statement, the assessment of your exemption will not hold up the peer review process.

The author’s answer: Thank you for informing us about the data sharing issue. In our study, the data is available at the following link: https://doi.org/10.7910/DVN/SQYQSY.

Q7: PLOS requires an ORCID iD for the corresponding author in Editorial Manager on papers submitted after December 6th, 2016. Please ensure that you have an ORCID iD and that it is validated in Editorial Manager. To do this, go to ‘Update my Information’ (in the upper left-hand corner of the main menu), and click on the Fetch/Validate link next to the ORCID field. This will take you to the ORCID site and allow you to create a new iD or authenticate a pre-existing iD in Editorial Manager. Please see the following video for instructions on linking an ORCID iD to your Editorial Manager account: https://www.youtube.com/watch?v=_xcclfuvtxQ.

The author’s answer: We apologize for not linking the ORCID iD. It has been provided in the revised submission.

Q8: Please include captions for your Supporting Information files at the end of your manuscript, and update any in-text citations to match accordingly. Please see our Supporting Information guidelines for more information: http://journals.plos.org/plosone/s/supporting-information. 

The author’s answer: We apologize for not adding the supporting information files at the end of the manuscript. We have included the supporting information in the revised version at the appropriate locations.

Q9: Please review your reference list to ensure that it is complete and correct. If you have cited papers that have been retracted, please include the rationale for doing so in the manuscript text, or remove these references and replace them with relevant current references. Any changes to the reference list should be mentioned in the rebuttal letter that accompanies your revised manuscript. If you need to cite a retracted article, indicate the article’s retracted status in the References list and also include a citation and full reference for the retraction notice.

The author’s answer: Thank you for your suggested revisions to the references. We have carefully reviewed and corrected the reference formatting, and we have provided explanations in the rebuttal letter.

SUGGESTIONS FROM REVIWERS

Reviewer #1

Title:

-The title is long and contains multiple complex terms, which might make it difficult for readers to quickly grasp the main focus of the study.

-The title does not specify what RUS is, which can be confusing for readers unfamiliar with the term.

-Terms like "5-HT2AR" and "DRD2" are highly specific and may not be immediately understood: Technical Jargon

The author’s answer: We are grateful for you suggestion, which have helped to clarify and refine the title. Following your advice, we have reduced the length of the title to make it clearer and have provided an explanation for Ruscogenin. We have also changed “5-HT2AR” and “DRD2” to “5-hydroxytryptamine” and “dopamine” respectively. The revised title is “Role of ruscogenin extracted from Radix Ophiopogon Japonicus in antagonizing 5-hydroxytryptamine and dopamine receptors through computational screening”.

Abstract:

-Q1: Abstracts should summarize the key findings without overwhelming the reader with technical specifics.

The author’s answer: Thank you for pointing out the issues with the abstract. We have made comprehensive and detailed changes according to your suggestions. 

We have expanded the description of the results. Line 11 has been changed from ‘In this study, 73 active ingredients of natural medicine extracts...’ to ‘To explore the potential effects of natural product extracts as therapeutic agents for psychiatric disorders, 73 active ingredients from natural medicine extracts were screened to identify potential inhibitors of the serotonin 2A receptor (5-HT2AR) and dopamine D2 receptor (DRD2) using computerized virtual molecular docking.’

Line 15 has been changed from ‘Then ADMET and molecular dynamics simulation analysis were utilized to investigate the drug-like property of ruscogenin and the mechanism of ruscogenin to antagonize DA2 and 5-HT2A receptors’ to ‘The most effective inhibitor of 5-HT2AR and DRD2 among these natural extracts was then evaluated for its drug-like properties using ADMET analysis, and its mechanisms of antagonism on DRD2 and 5-HT2AR were studied through molecular dynamics simulation.’

We removed unnecessary methodological techniques. Line 18 has been changed from ‘ADMET analysis showed that RUS possesses a favorable drug-like properties. Molecular dynamics simulation analysis indicated that RUS exhibits more stable RMSD plots...’ to ‘The results showed that ruscogenin (RUS) was the most effective inhibitor of 5-HT2AR and DRD2, possessing favorable drug-like properties (most values of ADMET analysis were within the optimal range).’.

-Q2: Some sentences are lengthy and complex, which can make the abstract harder to read.

The author’s answer: Thank you very much for your suggestions. We have modified the sentence to enhance its readability. 

Line 20 has been changed from ‘RUS exhibits more stable RMSD plots, lower RMSF values between residues 50-260, stronger hydrogen bonding interactions, higher compactness, smaller SASA value, and lower binding free energy in complex with 5-HT2A receptor in comparison to risperidone’ to ‘When compared to risperidone, RUS exhibited more stable root mean square deviation (RMSD) plots, lower root mean square fluctuation (RMSF) values from residues 50 to 260, stronger hydrogen bonding interactions, higher compactness, a smaller solvent-accessible surface area (SASA) value, and lower binding free energy (-43.81 kcal/mol vs. -35.68 kcal/mol).’

-Q3: While the abstract mentions that RUS showed favourable properties and stability, it does not quantify these results or provide specific data points, which could help in understanding the magnitude of the findings.

The author’s answer: Thank you for your valuable suggestions. According to the revision comments, we have added the presentation of specific results in line 19, line 23, and line 26. 

-Q4: The conclusion suggests that RUS is a promising candidate for therapeutic development based on the in-silico results. However, it would be prudent to temper this with a statement about the need for further validation through in vitro and in vivo studies.

The author’s answer: Thank you for your valuable opinions, which have enhanced the rigor of the paper. We have added the description of this part in line 27.

Introduction:

-Q1: The introduction covers a wide range of topics, from the general benefits of natural products to specific details about ruscogenin (RUS) and its potential therapeutic effects. While comprehensive, the flow can be confusing. A clearer structure with more defined sections would improve readability.

The author’s answer: Thank you very much for your constructive suggestions. Following your suggestions, we have relocated the discussion of ruscogenin (RUS) and its potential therapeutic effects to the ‘Discussion’ section. We have also expanded on the potential of natural products as antipsychotic agents, leading into a discussion on neurotransmitter receptors. This restructuring has clarified the presentation and improved the flow of the content.

-Q2: The introduction is quite long and contains a lot of detailed information. While detail is important, some of the more specific data (chemical formulas, specific biological activities) might be better suited for later sections.

The author’s answer: Thank you very much for your constructive suggestions. Following your suggestions, I have relocated the chemical formula for ruscogenin (RUS, C27H42O4), which was originally at line 32, to the ‘Results’ section. Additionally, the description of ruscogenin (RUS), a major bioactive steroidal sapogenin derived from the natural product radix ophiopogon japonicus, along with its various biological activities such as anti-inflammation, anti-tumor, anti-thrombosis, cardiac protection, and blood-brain barrier dysfunction, has been moved to the ‘Discussion’ section. Furthermore, the statement regarding the study that demonstrates ruscogenin’s ability to increase membrane fluidity has been placed at line 436 in the ‘Discussion’ section.

-Q3: Some sections, such as the detailed description of dopamine (DA) and serotonin (5-HT) receptors and their related pathologies, while informative, are quite extensive. This information might be better condensed to maintain focus on the main topic.

The author’s answer: Thank you very much for your constructive suggestions. We have condensed this section to maintain focus on the main topic. We merged the dopamine (DA) and serotonin (5-HT) receptors into one paragraph and removed unnecessary content.

-Q4: The introduction lacks sufficient citations to support the claims made, especially in the initial paragraphs discussing the effectiveness of natural products and the biological activities of RUS.

The author’s answer: Thank you for your feedback on the introduction. We appreciate your point regarding the lack of sufficient citations. To address this, we have expanded the discussion and incorporated additional references to support our claims about the effectiveness of natural products in lines 33-41. 

-Q5: Transitions between different sections could be smoother.

The author’s answer: Thank you for your valuable comments. We have made the necessary revisions. At the end of the first paragraph of the Introduction, we introduced the topic of neurotransmitters to seamlessly transition into the second paragraph. In the conclusion of the second paragraph, we have summarized the roles of DRD2 and 5-HT2AR as pharmaceutical targets. The third paragraph begins by revisiting these pharmaceutical targets and introduces the technical methods utilized in this study.

-Q6: The introduction does not clearly state the objective or hypothesis of the study until the very end. Introducing the aim of the research earlier would provide context for the detailed information that follows.

The author’s answer: We appreciate your valuable comments and have added the aim of the study at the end of the Introduction.

Methodology

-Q1: The specific version of Autodock Vina should be clarified. If it is 1.2.0, ensure the correct reference is provided.

The author’s answer: We appreciate your comments and have specified the version of AutoDock Vina used in our study. Additionally, we have verified the reference for version 1.2.0.

-Q2: Details on the parameters used for hydrogenation, charge calculation, and protonation should be more specific.

The author’s answer: We thank you for your comments and have included a detailed description in lines 93-96 of the manuscript. 

-Q3: The method for defining the binding pocket should be more explicitly described. Are specific residues or a particular region targeted?

The author’s answer: Thank you very much for your revision suggestions. We have expanded upon the specific description of the binding pocket in lines 96-98. For DRD2, binding pocket defined by the side chains of helices III, V and VI. For 5HT2AR, the bottom hydrophobic cleft was select as its bonding pocket. 

-Q4: Explanation needed on why semi-flexible docking was chosen and how flexibility was introduced.

The author’s answer: We appreciate your suggestion and have added an explanation at lines 99-103 detailing why semi-flexible docking was chosen and how flexibility was incorporated into the methodology.

-Q5: Justification for selecting leaprc.protein.ff99SB and leaprc.gaff force fields. Are there specific advantages for this study?

The author’s answer: We appreciate your comment and have expanded upon the description of the leaprc.protein.ff99SB and leaprc.gaff force fields at lines 119-127, and highlighting their respective advantages.

-Q6: The rationale behind choosing a 12 Å gap and the TIP3P water model should be provided.

The author’s answer: Thank you very much for your revision suggestions. In lines 126-129, we have detailed the rationale for employing the TIP3P water model and have included pertinent references regarding the choice of a 12 Å gap.

-Q7: Specify the concentration of Na+ or Cl- ions added.

The author’s answer: We especially appreciate your revision suggestions. We have provided a detailed description of the amounts of Na+ or Cl- ions added in lines 130-132.

-Q8: Expla

---

## [Editor Report · Decision Letter 1]

11 Sep 2024

Role of ruscogenin extracted from Radix Ophiopogon Japonicus  in antagonizing 5-hydroxytryptamine and dopamine receptors through computational screening

PONE-D-24-30011R1

Dear Dr. Yongmei Liu,

We’re pleased to inform you that your manuscript has been judged scientifically suitable for publication and will be formally accepted for publication once it meets all outstanding technical requirements.

Kind regards,

Akingbolabo Daniel Ogunlakin, Phd

Academic Editor

PLOS ONE
---

## [Editor Report · Acceptance letter]

16 Sep 2024

PONE-D-24-30011R1 

PLOS ONE

Dear Dr. Liu, 

I'm pleased to inform you that your manuscript has been deemed suitable for publication in PLOS ONE. Congratulations! Your manuscript is now being handed over to our production team.

Kind regards, 

on behalf of

Dr. Akingbolabo Daniel Ogunlakin 

Academic Editor

PLOS ONE